:PLOS | ONE

# Prosthesis design of animal models of periprosthetic joint infection following total knee arthroplasty: A systematic review

Ke Jie[1], Peng Deng[1,2], Houran Cao[1], Wenjun Feng[2], Jinlun Chen[2], Yirong Zeng[2]*

**1** The First Clinical Medical College, Guangzhou University of Chinese Medicine, Baiyun District, Guangzhou, Guangdong Province, China, **2** The Third Department of Orthopedics, The First Affiliated Hospital of Guangzhou University of Chinese Medicine, Baiyun District, Guangzhou, Guangdong Province, China

* zengryirong1966@126.com

**Data Availability Statement:** The data can be found in Pubmed, EMbase, Cochrane Library, Web of Science, Wanfang Data and China National Knowledge Infrastructure.

## Abstract

### Background

The number of periprosthetic joint infections (PJI) after total knee arthroplasty (TKA) is increasing annually. Animal models have been used to clarify their clinical characteristics and the infection mechanism of pathogenic bacteria, However, since the prosthesis design of animal models is not uniform, it is difficult to simulate the environment of clinical PJI.

### Objectives

To retrospect the progress on the prosthesis design of animal models of PJI after TKA and to summarize the criteria for evaluating a clinically representative model of PJI.

### Methods

This systematic review was reported on the basis of Systematic Reviews and Meta-Analyzes (PRISMA). Pubmed, EMbase, Cochrane Library, Web of Science, Wanfang Data and China National Knowledge Infrastructure were researched for animal models of PJI after TKA from database establishment to April 2019 according to Chinese and English retrieval words, including "periprosthetic joint infections and total knee arthroplasty," "periprosthetic joint infections and model," "periprosthetic joint infections and biofilm," and "total knee arthroplasty and model."

### Results

A total of 12 quantitative studies were enrolled in our study finally: 8 representative studies described prosthesis designs used in PJI animal models, 4 studies described prosthesis designs in non-infected animal models which were suitable for infection models. The major problems need to be dealt with were prosthesis, installation location, material, the function of separating the articular and medullary cavity, fixation manner, and the procedure of preserving the posterior cruciate ligament.

**Funding:** The current study received High-Level Hospital Construction Project of The First Affiliated Hospital of Guangzhou University of Chinese Medicine (Grant number 211010010121).

**Competing interests:** The authors have declared that no competing interests exist.

## Conclusion

A highly representative design of the animal prosthesis of PJI should meet the following criteria: the surface of the prosthesis is smooth with the formation of biofilm, composed of titanium-6Al-4V or cobalt-chromium-molybdenum alloy; prosthesis can bear weight and is highly stable; and it can connect the joint cavity and medullary cavity simultaneously. To reach a more reliable conclusion, further experiments and improvements are required.

## Introduction

Periprosthetic joint infection (PJI) is one of the severe complications of total knee arthroplasty (TKA), accounting for 25%-38% of postoperative complications in TKA [1–4]. Although through a series of measures [5–8] (including the use of perioperative antibiotics, intraoperative antibiotic cement, intraoperative antibiotic calcium sulfate, improvement of surgical environment, and elimination of local bacterial colonization), the infection rate has been controlled at about 1%-3% [9, 10]. However, with the rapid growth of the population undergoing TKA in recent years, the number of people with PJI has also increased annually [11, 12]. Two-stage revision, involving an antibiotic-impregnated polymethylmethacrylate spacer, is thought to be the gold standard for treating PJI, but the current cure rate for two-stage revision is only 72%-95% according to several studies [7, 11, 13].

To better understand and recognize the diagnosis of PJI and develop novel treatment strategies, it is important to investigate its clinical characteristics and infection mechanism of pathogenic bacteria. Nowadays, many studies focus on in vivo and in vitro experiments. However, conclusions based on in vitro experiments do not guarantee the authenticity of the infection mechanism and the clinical effectiveness of various interventions. On the other hand, experiments in animals are also not able to completely simulate the clinical pathogenesis of PJI and transform our understanding of PJI due to synthetic factors, among which the prosthesis design of animal models acts as an important medium. Its clinical pathogenesis has been widely discussed in the past [14–22], which can help us better understand the colonization of bacteria, formation of biofilm, and adhesion process and resistance to the host immune response [23, 24]. The implant design was the first step of performing PJI models, if it is far different from clinical implant, then the next steps will be affected and the results may be lack of persuasion. Nevertheless, the current design of PJI models is still dissatisfying. They are not clinically representative and can barely reproduce the periprosthetic environment because of the lack of weight bearing, poor matching, only partial replacement, weak stability, inconsistent relevantly clinical material, rough surface, and non-anatomical appearance [14, 16, 25–30]. According to the International Consensus on Orthopedic Infections in 2019, the ideal prosthesis design of a PJI model has yet to be established [31].

In this review, we retrospectively collected information about previously established prostheses used in PJI animal models of knee replacement, and herein, we discuss their advantages and disadvantages in order to summarize the most clinically relevant and reproducible evaluation criteria of prostheses, and to promote the development of a novel animal prosthesis. Thus, our review may better a provide theoretical basis for clinical diagnosis, treatment, and prevention of PJI.

## Methods

### Data sources and search strategy

This systematic review was reported on the basis of Systematic Reviews and Meta-Analyzes (PRISMA) statements for prosthesis design of animal models of PJI following TKA [32].

One investigator (PD) design the search strategy and two investigators (WJF and JLC) completed the literature search independently. After cross-checking, the disagreements over the included articles were submitted to the third person (HRC) for arbitration. Studies associated with PJI models were identified using Pubmed, EMbase, Cochrane Library, Web of Science, Wanfang Data and China National Knowledge Infrastructure from database establishment to April 1st 2019 using the following keywords: (1) "periprosthetic joint infections and total knee arthroplasty," (2) "periprosthetic joint Infections and model," (3) "periprosthetic joint infections and biofilm," and (4)"total knee arthroplasty and model." In addition to electronic retrieval, all the relative references in the included studies were searched manually to avoid omitting undiscovered studies in the retrieval database.

## Study selection and eligibility criteria

Two investigators (WJF and JLC) read the titles and abstracts of all downloaded literature, then preliminarily excluded the research that obviously did not meet the eligibility criteria, and finally carefully scanned the full text to screen out the studies that met the standards for data extraction. The eligibility criteria in this study were: (1) animal trials, (2) studies describing content of prosthesis design of PJI following TKA in detail, (3) the prosthesis that was clinically representative or used frequently by different researchers in the past or recently. The exclusion criteria were as follows: (1) Literature reviews, conference abstracts, letters to the editor, (2) studies describing prosthesis too briefly, (3) Repetitive prosthesis design, (4) Non-animal trial.

## Data extraction and items

After scanning the database, we analyzed their advantages and disadvantages on the prosthesis design of animal models of PJI, including author, publication years, experimental subject, prosthesis and their characteristics(location, material, whether it was located in the weight-bearing area or not, separate the articular and medullary cavity or not), operative procedure (whether cement was used or not, PCL was preserved or not), inoculation bacteria and their details(species, inoculation location, dose and concentration). These are the international issues at present, which will be answered with'yes', 'no', 'not mention' or matching descriptions in **Tables 1 and 2**.

## Risk of bias within studies

The STAIR [33] (the initial Stroke Therapy Academic Industry Roundtable) risk of bias tool was used to independently assess the methodological and reporting quality of included studies by two investigators (WJF and JLC), and the divergences were submitted to the third person (HRC) for arbitration after checking each other. The included studies were assessed across the following factors: sample size calculation, inclusion and exclusion criteria, randomization, allocation concealment, reporting of animals excluded from analysis, blinded assessment of outcome, reporting potential conflicts of interest and study funding. Finally, the quality score was calculated according to above information. The total scores were 7 points, and a study of more than or equal to 3 points were defined as high quality research.

## Results

### Search results

A total of 4299 records were searched through English and Chinese database. After removing duplicates and screening topics and abstracts, 481 records remained. Then the full text and

**Table 1. Animal models of prosthetic joint infection.**

| Author, Year | Animal | Prosthesis | | | | Preserving PCL or not | Bacteria | | |
|---|---|---|---|---|---|---|---|---|---|
| | | Implant, Location, Material | Located in the weight-bearing area or not | Separating the articular and medullary cavity or not | Using cement or not | | Species | Inoculation location | Dose, concentration (cfu/ml) |
| Bernthal 2010[14] | Mouse | Femoral medullary cavity, stainless steel K-wire | No | No | No | Yes | *S.aureus* | Knee cavity | 2ul, $5 \times 10^2$ to $10^4$ |
| Carli 2017[15] | Mouse | Tibial component, tantalum | Yes | Yes | No | Yes | *S.aureus* | Knee cavity | 2ul, $3 \times 10^5$ |
| Craig 2005[16] | Rabbit | Lateral femoral condyle, stainless steel hollow nail + UHMWPE insert | No | No | Yes | Yes | *MRSA* | Knee cavity | 0.1ml, $1 \times 10^2$ to $10^3$ |
| Saleh-Mghir 2011[30] | Rabbit | Tibial component, silicone | Yes | Yes | NM | NM | *MRSA* | Knee cavity | 0.5ml, $5 \times 10^7$ |
| Poultsides 2010[34] | Rabbit | ①Tibial medullary cavity, Cylinder, Tantalum ②Proximal tibia, circular silicone | Yes | Yes | NM | NM | *MRSA* | Femoral artery | 1ml, $3-5 \times 10^8$ |
| Kalteis 2006[37] | Rat | Femoral medullary cavity, hollow nail(NM materials) | No | No | No | NM | *S.aureus* | Femoral medullary cavity | 100μL, $1 \times 10^8$ |
| Petty 1985[38] | Dog | Femoral medullary cavity, Cylinder,①stainless steel, ②cobalt-chromium alloy ③polymer polyethylene ④cement | No | YES | ①②③NO, ④Yes | NM | *S.aureus, S. epidermidis E. coli* | Femoral medullary cavity | $1 \times 10^2$ to $10^8$ |
| Schurman 1975[39] | Rabbit | Suprapatellar bursa, stainless steel particles | No | No | - | - | *S.aureus* | Suprapatellar bursa | - |

**NOTE:** *MRSA = methicillin-resistant staphylococcus aureus, S.aureus = Staphylococcus aureus*, UHMWPE = Ultra-High Molecular Weight Polyethylene

*S.Epidermidis = Staphylococcus epidermidis, E. Coli = Escherichia coli*, K-wire = Kirschner wire, PCL = Posterior Cruciate Ligament, NM = not mention

their references were carefully read and analyzed. According to the eligibility and exclusion criteria, 12 articles were finally selected, 8 of which representative studies described prosthesis designs used in PJI animal models and 4 described prosthesis designs in non-infected animal models. The PRISMA flow diagram was presented in detail in **Fig 1**.

## Assessment of methodological and reporting quality

Based on the STAIR tool, all the details of quality of 12 studies were presented in **Table 3**. 6 articles were assessed as high quality [16, 22, 30, 34–36]. All 12 studies referred to potential conflicts of interest and study funding [14–16, 19, 22, 30, 34–39], while no studies reported the sample size calculation. Over a half of the included studies (58%, 7/12) reported animals excluded from analysis [16, 22, 30, 34–36, 39]. In 42% (5/12) of the included studies, randomization of the experiment was reported [16, 30, 35–37]. Only 8% (1/12) of the included studies described the inclusion and exclusion criteria [22], and 17% (2/12) described allocation concealment [16, 35], and 17% (2/12) described blinded assessment of outcome [34, 35].

**Table 2. Non-infective animal models.**

| Author, Year | Animal | Prosthesis | | | | Preserving PCL or not |
|---|---|---|---|---|---|---|
| | | Implant, Location, Material | Located in the weight-bearing area or not | separating the intra-articular and medullary cavity or not | Using cement or not | |
| Turner 1989[19] | Dog | Femoral component, cobalt-chromium alloy<br>Tibial component, tantalum alloy+50% dense fibrous metal plate<br>Insert, HMWPE | Yes | Yes | Yes | Yes |
| Yan Zhi Qiang 2014[22] | Rabbit | Femoral component, Co-Cr-Mo alloy<br>②Tibial component, UHMWPE | Yes | Yes | Yes | Yes |
| Xu Yang 2015[35] | Mouse | Tibial component, Tantalum alloy | Yes | Yes | No | Yes |
| Zampelis 2013[36] | Rabbit | Tibial component, NM materials | Yes | Yes | No | NM |

**NOTE:** UHMWPE = Ultra-High Molecular Weight Polyethylene, HMWPE = High Molecular Weight Polyethylene, PCL = Posterior Cruciate Ligament, NM = not mention

## Study characteristics

**The progress of prosthesis designs used in PJI animal models.** The animal model is a reasonable and efficient approach to transform all kinds of results in in vitro into the clinical setting. At present, there is great controversy regarding the design of prostheses, and various prostheses were used in PJI animal models [14–16, 30, 34, 37–39], which may cause confusion among researchers. These studies chose the most classic, representative, and widely used prostheses to analyze their characteristics and explore the developmental direction of new models in the future. Details are shown in **Table 1**.

The PJI animal model was first proposed and designed by Schurman in 1975 [39]. In order to evaluate the susceptibility of prostheses to PJI, stainless steel particles and *Staphylococcus aureus* were implanted into a rabbit's suprapatellar bursa. The prosthesis was suspended in saline rather than implanted in the bone so it did not simulate the bone-cement-prosthesis interface during TKA.

In 1985, Petty et al [38] recognized that there may be some correlations between the different implant materials and infection of pathogenic bacteria. After injecting different kinds of bacteria, including *S. aureus*, *S. epidermidis*, and *Escherichia coli*, into the medullary cavity at the distal end of the femur in dogs, different cylinders, composed of stainless steel alloy, cobalt-chromium alloy, high-density polyethylene, polymerized polymethylmethacrylate, or polymethylmethacrylate, respectively, were then introduced. Although larger animals like dogs have musculoskeletal and immunological systems similar to humans and can preferably mimic the environment of the human knee joint, their use will always be accompanied by more ethical challenges, financial costs, and lower throughput [25]. Recently, researchers had preferred to choose small animals, such as rabbits, rats, and mice, in PJI models. In 2010, Bernthal et al [14] used the same method in that a stainless steel Kirschner wire was retro-gradely injected into the distal femur of mice for PJI modeling (**Fig 2A**). The implant in the medullary cavity was originally designed to imitate PJI after TKA, but in fact, it might confuse osteomyelitis models with PJI models [40, 41]. Because of its simplicity and reproducibility of manipulation, many researchers still used this method for PJI modeling [17, 18, 20, 21, 42].

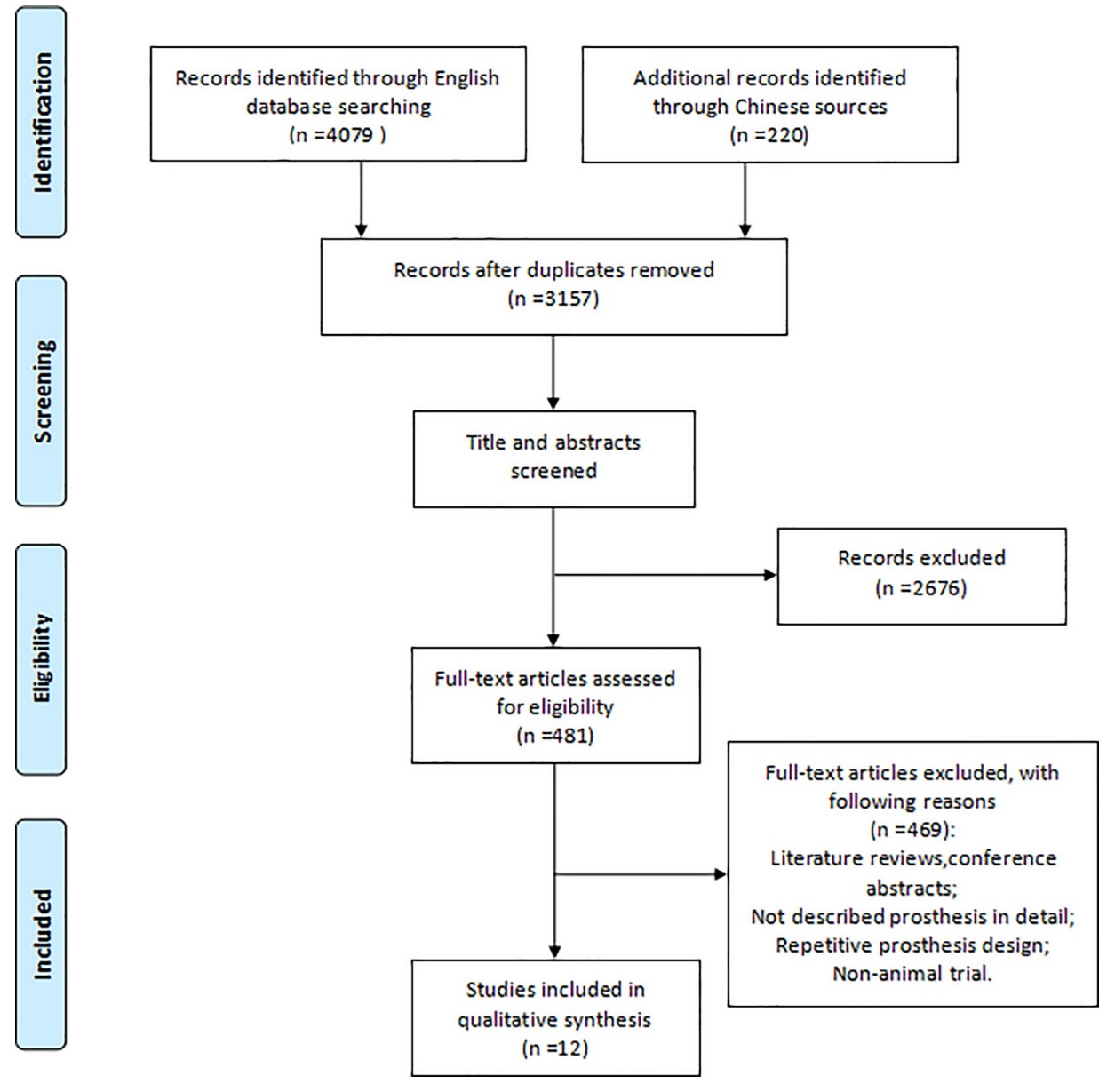

**Fig 1. Study screening flow.**

However, the prosthesis had the following obvious defects. Only stainless steel materials were used, which were different from the current commonly titanium (Ti)-6Al-4V and cobalt-chromium-molybdenum alloy in TKA. In addition, it can not achieve the load-bearing state, and the authors did not use bone cement to fix the prosthesis. Although the minimal infecting dose was low, it cannot reproduce the periprosthetic environment. All these factors affected the formation of biofilm on the surface of the prosthesis [43]. Bernthal et al [14] also suggested that the cartilage in femur and tibia, which were not removed intraoperatively, may interact with the pathogenic bacteria.

In another study reported by Craig et al in 2005 [16], after drilling a hole in the lateral femoral condyle anterior to the lateral collateral ligament, 0.1-ml bone cement, a fully threaded

**Table 3. Risk of bias.**

| Study | Sample size calculation | Inclusion and exclusion criteria | Randomization | Allocation concealment | Reporting of animals excluded from analysis | Blinded assessment of outcome | Reporting potential conflicts of interest and study funding | Quality score |
|---|---|---|---|---|---|---|---|---|
| Bernthal 2010[14] | N/A | N/A | N/A | N/A | N/A | N/A | M | 1 |
| Carli 2017[15] | N/A | N/A | N/A | N/A | N/A | N/A | M | 1 |
| Craig 2005[16] | N/A | N/A | M | M | M | N/A | M | 4 |
| Turner 1989[19] | N/A | N/A | N/A | N/A | N/A | N/A | M | 1 |
| Yan Zhi Qiang 2014[22] | N/A | M | N/A | N/A | M | N/A | M | 3 |
| Saleh-Mghir 2011[30] | N/A | N/A | M | N/A | M | N/A | M | 3 |
| Poultsides 2008[34] | N/A | N/A | N/A | N/A | M | M | M | 3 |
| Xu Yang 2015[35] | N/A | N/A | M | M | M | M | M | 5 |
| Zampelis 2013[36] | N/A | N/A | M | N/A | M | N/A | M | 3 |
| Kalteis 2006[37] | N/A | N/A | M | N/A | N/A | N/A | M | 2 |
| Petty 1985[38] | N/A | N/A | N/A | N/A | N/A | N/A | M | 1 |
| Schurman 1975[39] | N/A | N/A | N/A | N/A | M | N/A | M | 2 |

**NOTE:** N/A = not available; M = mentioned.

stainless steel hollow nail, and ultra-high molecular weight polyethylene (UHMWPE) washer were implanted successively (**Fig 2B**). The three materials used in this study were closer in similarity to TKA materials used in clinical practice, and they had increased stability; thus, they were also favored by many researchers [44–46]. In terms of anatomical location, the materials were located on the outside of the knee joint so that they can bear only compressive stress in the vertical direction, not rotational stress. Furthermore, the materials did not separate the medullary cavity and joint cavity, and no articular cartilage was removed intraoperatively. All these factors limited their further application in the future.

In 2017, Carli et al [15] first applied a three-dimensional (3D) printed tibial prosthesis with the Ti-6Al-4V in a PJI mice model (**Fig 2C**). This method was a great breakthrough in the development of the prosthesis design. Initially, this prosthesis originated from a non-infected model, and was first proposed and applied by Xu Yang et al in 2015 [35]. In order to assess the effect of recombinant human parathyroid hormone on cancellous bone integration, they developed an uncemented tibial prosthesis for a mouse model, whose surface was rough (**Fig 2D**). This prosthesis combined the advantages of several prostheses aforementioned [14, 16, 38, 39], including the ability to bear weight, separate the intra-articular and intramedullary cavities, use clinically relevant materials, and require simple manipulation. It was more similar to the tibial component replacement in clinical TKA and more representative than other prostheses of PJI. The former also met four criteria of clinically representative PJI models proposed by Carli et al in 2016 [25]. The criteria included the following conditions: (1) Biofilm can be

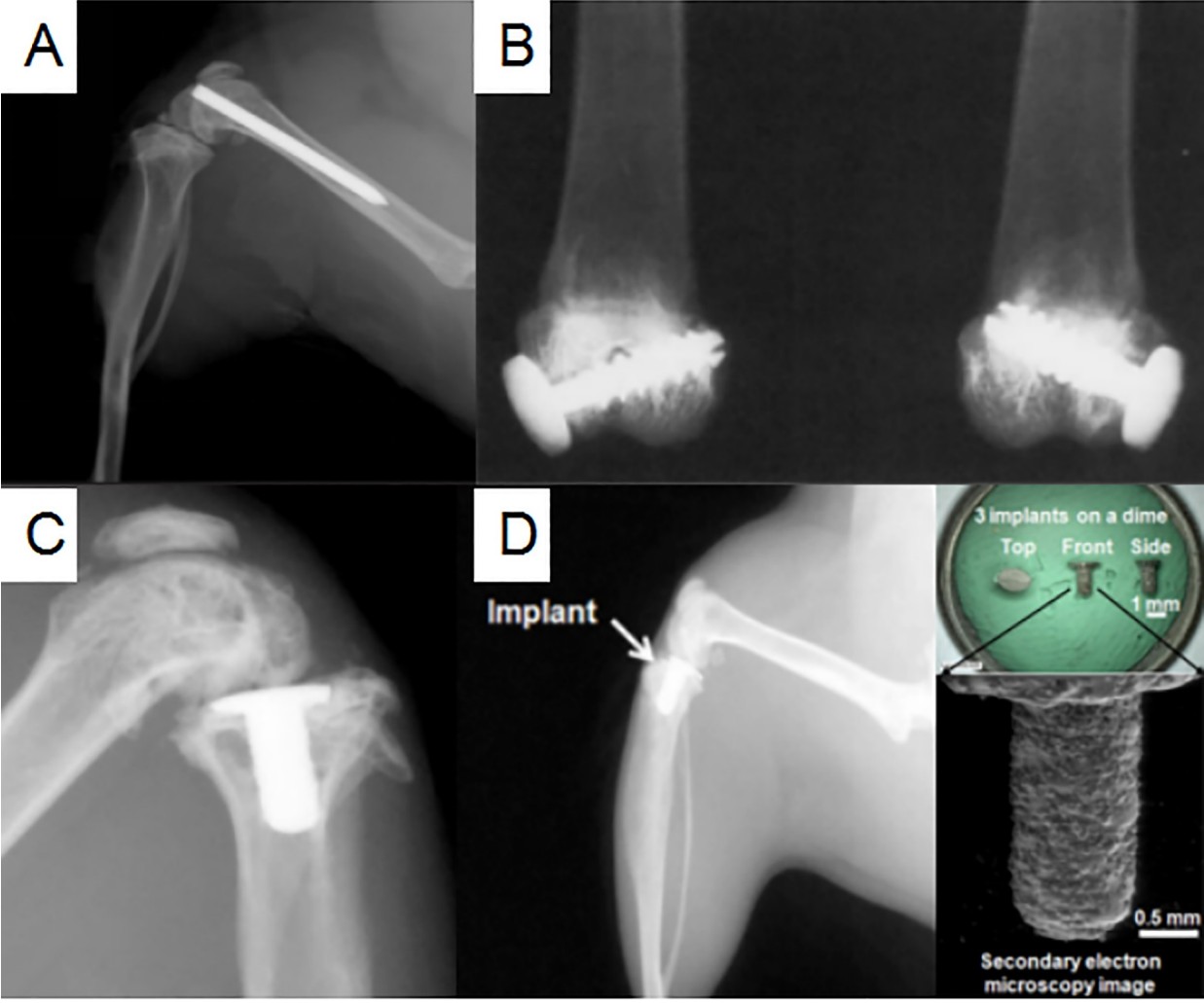

**Fig 2. The design of four prostheses. (The figures were derived from references.). (A)** A stainless steel Kirschner wire retrogradely inserted into the distal femur of a mouse model [14].(**B**) A full threaded stainless steel hollow nail and UHMWPE washer implanted in the lateral femoral condyle and anterior to the lateral collateral ligament of a rabbit model [16]. (**C**) A 3-dimensionally printed Ti-6Al-4V prosthesis implanted in the tibial plateau of a mouse model [15].(**D**) A 3-dimensional printed prosthesis implanted in the tibia of a murine model [35].

formed on the surface of the prosthesis; (2) Prosthetic materials should be similar to clinical materials, bear weight, and create an intra-articular environment; (3) The animals chosen for models should have musculoskeletal and immunological system compared to human beings; (4) The bacteria, biofilm, and host immune response can be measured quantitatively. Among them, the first and second sections were aimed at the prosthesis design. In fact, in addition to the use of uncemented fixation and unknown stability, only tibial replacement had been performed without femoral replacement, which was also one of the problems that needs improved in the future.

**Designing PJI models based on non-infected animal models.** As mentioned above, the design of the PJI animal model has made great progress, but there are still some differences from PJI after TKA in the clinical setting. How to improve the prosthesis is still the focus. With the advent of the age of multidisciplinary communication, some prostheses in non-infection models may be more suitable for infection models, which can accommodate for the

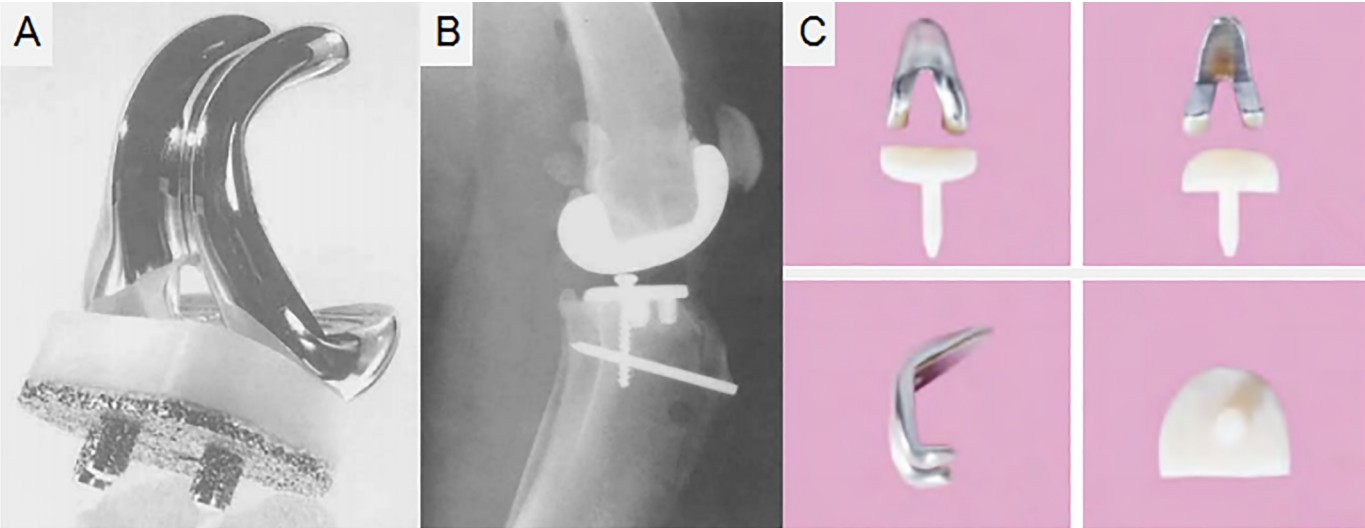

**Fig 3. The appearance of two prostheses. (The figures were derived from references.) (A-B)** An unrestricted posterior CR prosthesis implanted in a dog model [19].**(C)** An anatomical joint prosthesis implanted in a rabbit model [22].

shortcoming of PJI models and make the prostheses closer to perfect [15, 19, 35, 36]. Details are shown in **Table 2**.

As early as 1989, Turner et al [19] designed an unrestricted posterior cruciate-retaining (CR) prosthesis for dogs to observe bone ingrowth of the tibial component in TKA. The posterior cruciate ligament was preserved as much as possible, and patella replacement was not performed during the operation (**Fig 3A and 3B**). This prosthesis consisted of three parts: the femoral prosthesis, insert, and tibial prosthesis similar to that used in clinical TKA. In terms of appearance and material science, the femoral prosthesis was made of a cobalt-chromium alloy, imitating the contour of the distal femur of dogs, and the insert was comprised of UHMWPE. The tibial component was made of titanium alloy, below which was a 50% dense fibrous metal plate and three cylindrical pegs coated with fiber metal, above which was the UHMWPE. In terms of stability, the femoral part was fixed with bone cement, tibial part was fixed by bone ingrowth, and a screw was inserted into the cancellous bone behind the tibial prosthesis to increase the stability. Regarding the degree of matching, the femoral anterior condyle, posterior condyle, and distal condyle underwent osteotomy with alignment and cutting saws, and the tibial posterior slope angle (TPSA) was maintained at about 25˚. Moreover, a femoral trocar groove was designed in the femoral prosthesis to ensure matching with the patellofemoral joint after joint replacement.

Six months postoperatively in Turner et al's study [19], the x-ray showed no changes in the position of the tibial or femoral prosthesis, and only a small amount of osteophytes was seen in the anterior, medial, and lateral sides of the tibial prosthesis. Unexpectedly, although cartilage wear and even subchondral invasion can be seen in the patellofemoral joint, it was stable. The prosthesis can be used for reference in the design of the PJI animal model, because it showed great advantages in implant stability, load-bearing, clinically relevant materials, which met the requirements for PJI models. However, it did not separate the intra-articular and intramedullary spaces. Furthermore, compared to small animals, dogs, sheep, and other large animals have simpler joint exposure, larger operating space, and relatively easier prosthesis installation, and the manipulation has less of an effect on postoperative function. However, it may be

extremely difficult to apply the prosthesis to the knee joint of small animals. Because of the cross-domain application, the details need to be further adjusted in the future.

Not only in the field of infection, but also in other fields, the TKA prostheses of animal models are not uniform [19, 22, 36]. In order to provide standardized multi-model prostheses, in 2014, Yan et al [22] designed an anatomical joint prosthesis for rabbits (**Fig 3C**). First, computed tomography was used to scan the knee joints of rabbits with different weights and knee prostheses of humans. Then 3D reconstruction was performed to reduce the equal proportion of the prostheses of humans and adjust it according to the actual anatomic markers of the rabbit knee joint, including the internal-external and anteroposterior diameters of the femoral condyles and tibial plateau, diameters of the femoral and tibial medullary cavities, and so on. The tibial component was made of UHMWPE, and the femoral component was made of Co-Cr-Mo alloy. Cement was used for fixation. The results showed that the prosthesis was found to be in a good position, and no obvious loosening was found at 1 month postoperatively. The rate of excellent function postoperatively was 87% (13/15) within 7 days, flexion and extension of the knee joint on the operated side was more tense than that on the normal side, and 13 rabbits returned to normal crawling 7 days postoperatively. The design of the prosthesis in small animals is more challenging than that in large animals. It is not only difficult to expose the bone during operation, but more precise osteotomy and a higher degree of prosthesis matching are required. Although the authors used a fully anatomical prosthesis and did not install an insert, the knee joint on the operated side was still tight so as to affect their gait. The tibial component may be too thick to allow the knee joint activity. Eventually, 2 rabbits developed postoperative dysfunction. Other problems also existed, for example, the femoral component did not separate the intra-articular and intramedullary spaces, and it may demand higher surgical technique.

## Discussion

### Question 1: Should bone cement be used for fixation?

The achievement of implant stability plays an important part in the formation of bacterial biofilm and periprosthetic immune response. The instability of the prosthesis will not only affect the bone growth, resulting in loosening of the prosthesis [47], but it will also affect the adhesion of bacteria due to the unstable shear force [26]. In clinical practice, the application of cementless TKA is limited because it may be associated with higher loosening and revision rates than cemented TKA for lower initial stability [48]. Although cementless fixation is reliable according to many studies [49, 50], it may not be determinable by radiography whether the prosthesis has fretting within a short follow-up period. Furthermore, it has been reported that polymethylmethacrylate is also a good material for biofilm attachment [51, 52]. Other studies have shown that biofilm can even form on an antibiotic-loaded bone cement [53, 54], which is closely linked with the progression of PJI.

### Question 2: Should one resect or preserve the posterior cruciate ligament?

The anatomical structure of dogs, rabbits, rats, and mice are similar to humans, and their knee joints are commonly selected as objects in experimental studies [14–20], because they have both anterior and posterior cruciate ligaments, medial and lateral collateral ligaments, and medial and lateral menisci. On the other hand, because of the obvious difference in walking gait, the anatomical characteristics of the lower extremity and limb alignment are different from those of humans. In theory, their hip-knee-ankle angle and TPSA in the sagittal position are much larger than those in humans. Relevant studies showed that the TPSA of dogs is about 23.6˚-27.4˚[55, 56], knee can reach about 138.5˚ in the standing position [57], and range of

motion (ROM) is about 120° [58]. Whenever dogs walk or stand, the knee is usually at the position of flexion, and the posterior cruciate ligament is very important for maintaining stability of their knees [59].

In clinical practice, according to whether the posterior cruciate ligament is preserved, there are two kinds of prostheses, including posterior CR prostheses and posterior cruciate-stabilized prosthesis (PS) [60, 61]. The CR prosthesis can better simulate the rolling mechanism and biomechanical characteristics of the normal knee, and achieve higher proprioceptive sensation [28]. The PS prosthesis takes the place of the function of the posterior cruciate ligament, by creating an upright between the center of the tibial component and a cam between the posterior condyle of the femoral component, which may not fully restore the function of the intact posterior collateral ligament, especially in deep-flexion activities. Additionally, unlike the PS prosthesis for humans, the design of the PS prosthesis for animal models requires higher accuracy; thus, the CR prosthesis is more suitable than the PS prosthesis for animal models of deep knee flexion. Therefore, when designing the appearance of an animal prosthesis, not only the anatomical features of the bone structure but also the functional characteristics should be considered carefully.

## Question 3: Should only tibial replacement be performed or should tibial and femoral replacement be performed simultaneously?

In a study by Carli et al [15], a stable biofilm could be formed after tibial plateau replacement following the injection of bacteria. Several studies have also shown that it is clinically representative [31, 53]. However, it does not satisfy the concept of the total anatomical prosthesis, which may result in a different periprosthetic environment while undergoing tibial replacement only or total condylar replacement, so it is difficult to meet the needs of basic clinical research. Although evidence is lacking, there may also be interactions between the preserved femoral articular cartilage and bacteria[14]. In contrast, performing femoral replacement simultaneously in animals will greatly increase the difficulty of surgery, which may lead to an increase of intraoperative bleeding, postoperative dysfunction, and so on [22]. Further experimental proof is needed to determine whether there is any difference between the two kinds of methods established in the PJI model.

## Question 4: Should a UHMWPE insert be used or not?

Karbysheva et al [62] suggested that the microorganisms' ability to adhere and form a biofilm on different biomaterials of explanted joint prosthesis components might differs among biomaterials. The implant components in 40 patients diagnosed with PJI were retrieved to perform sonication cultures, which were analyzed qualitatively and quantitatively. The results demonstrated that the bacteria counts were larger in the polyethylene group than in the titanium alloy and the cobalt-chromium alloy groups, which indicated that the polyethylene implant had higher microbial adhesion affinity in vivo. Another factor that needs to be considered carefully is whether a UHMWPE insert will affect joint ROM. For large animals, it has been reported that the UHMWPE insert can be used in the process of TKA [19]. This technique is evolved, and there is no obvious limitation in flexion and extension of the knee joint after implanting the UHMWPE insert. However, for small animals, the tibiofemoral joint space may be too narrow to accommodate for such an insert. Even if the insert can be successfully implanted, there is still a high risk of low ROM. Presently, there are no reports on the implantation of a polyethylene insert during TKA in a small animal model.

## Strengths and limitations

This is the first study concerning to the principles of implant design of the PJI models specially. We have not only systematically discussed implant stability, load-bearing, clinically relevant materials, separation of intra-articular and intramedullary spaces, but also the problems of fixation, posterior cruciate ligament, femoral replacement and UHMWPE insert.

Accompanied by evident strengths, several limitations also exist in this study. This study only focused on prosthesis design without other factors, however, a PJI model may be influenced by various complex factors, including the prosthesis design, selection of animal species, pathogenic bacteria, amount of bacteria implanted, immune environment around the prosthesis, method of implanting bacteria, and so on (**Table 1**), which will synthetically affect the density and quality of the biomembrane formed on the surface of the prosthesis. The reason why we chose this topic was that most influencing factors were based on the prosthesis design. Only by imitating the periprosthetic environment in the human body to the greatest extent can we create the most clinically representative model of PJI. Furthermore, the evaluation criterias recommended by us were established according to previous articles, so further experimental verification is required.

## Overview

PJI following TKA will cause patients to suffer from enormous physical injury and financial burden, and it will present new severe challenges for the medical staff. The establishment of a PJI animal model has become one of the important ways to overcome this difficult problem. There is still no consensus on the criteria for these factors because of the individual modeling methods and concept resulting in bias in the studies' results.Some achievements have been made in the design, but there are still many deficiencies; for example, the achievements can not be applied to clarify the clinical pathogenesis or to test the efficacy of drugs effectively. To reach a more reliable conclusion, the implant design needs to be improved and perfected.

## Conclusion

In conclusion, when evaluating the high clinical representativeness of a prosthesis design of animal models of PJI following TKA, we recommend the following evaluation criterias, of which the first six are relatively more important:

1. The surface of the prosthesis should be smooth so as not to limit knee joint ROM.

2. The surface of the prosthesis should be found the formation of biofilm.

3. The implants are composed of Ti-6Al-4V or Co-Cr-Mo alloy, with or without UHMWPE, which are similar to clinical materials.

4. The implants can bear weight and separate the intra-articular and medullary cavities for reproducing the periprosthetic environment.

5. Since high stability is needed, such as the use of bone cement, the bone-cement-prosthesis interface can reduce the impact of wear particles on bone resorption.

6. Tibial replacement and femoral prosthesis replacement can be performed simultaneously as possible.

7. The posterior cruciate ligament can be preserved as much as possible to increase the stability of the knee. In other words, the femoral and tibial prostheses can accommodate the end point of the posterior cruciate ligament.

8. If allowed, good patella track and patellofemoral joint matching can be achieved.

9. Manipulation is simple and reproducible. Therefore, patella surface replacement should not be performed, as there is no evidence to support it.

## Supporting information

**S1 File. PRISMA 2009 checklist.**
(DOC)

**S2 File. Search strategy.** The search strategies in each database and the inclusion and exclusion of articles.
(DOC)

## Author Contributions

**Conceptualization:** Ke Jie, Yirong Zeng.

**Data curation:** Peng Deng, Houran Cao, Wenjun Feng, Jinlun Chen.

**Investigation:** Houran Cao, Wenjun Feng, Jinlun Chen.

**Resources:** Houran Cao, Wenjun Feng, Jinlun Chen.

**Supervision:** Yirong Zeng.

**Writing – review & editing:** Ke Jie, Peng Deng.

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
