## [Decision Letter · Decision Letter 0]

30 Aug 2019

[EXSCINDED]

PONE-D-19-17399

Prosthesis Design of Animal Models of Periprosthetic Joint Infection Following Total Knee Arthroplasty: A Systematic Review

PLOS ONE

Dear Dr Zeng,

Thank you for submitting your manuscript to PLOS ONE. After careful consideration, we feel that it has merit but does not fully meet PLOS ONE’s publication criteria as it currently stands. Therefore, we invite you to submit a revised version of the manuscript that addresses the points raised during the review process.

We would appreciate receiving your revised manuscript by Oct 14 2019 11:59PM. To enhance the reproducibility of your results, we recommend that if applicable you deposit your laboratory protocols in protocols.io, where a protocol can be assigned its own identifier (DOI) such that it can be cited independently in the future. For instructions see: http://journals.plos.org/plosone/s/submission-guidelines#loc-laboratory-protocols

We look forward to receiving your revised manuscript.

Kind regards,

Daniel Pérez-Prieto, PhD

Academic Editor

PLOS ONE

Journal Requirements:

2. Please confirm that you have included all items recommended in the PRISMA checklist including details of reasons for study exclusions in the PRISMA flowchart and number of studies excluded for each reason.

4. a) Please provide an amended Funding Statement that declares *all* the funding or sources of support received during this specific study (whether external or internal to your organization) as detailed online in our guide for authors at http://journals.plos.org/plosone/s/submit-now.  

b) Please state what role the funders took in the study.  If any authors received a salary from any of your funders, please state which authors and which funder. If the funders had no role, please state: "The funders had no role in study design, data collection and analysis, decision to publish, or preparation of the manuscript."

7. Please ensure that you refer to Figure 3 in your text as, if accepted, production will need this reference to link the reader to the figure.

Reviewers' comments:

Reviewer's Responses to Questions

**Comments to the Author**

1. Is the manuscript technically sound, and do the data support the conclusions?

Reviewer #1: Yes

Reviewer #2: Yes

Reviewer #3: Yes

2. Has the statistical analysis been performed appropriately and rigorously? 

Reviewer #1: I Don't Know

Reviewer #2: Yes

Reviewer #3: N/A

3. Have the authors made all data underlying the findings in their manuscript fully available?

Reviewer #1: Yes

Reviewer #2: Yes

Reviewer #3: Yes

4. Is the manuscript presented in an intelligible fashion and written in standard English?

Reviewer #1: No

Reviewer #2: Yes

Reviewer #3: Yes

5. Review Comments to the Author

Reviewer #1: Current animal models of PJI are limited in their translational nature primarily because of their inability to recreate the periprosthetic environment. A clinically representative PJI model must involve an implant that recreates the periprosthetic space and be amenable to methodologies that identify implant biofilm as well as quantify the peri-implant bacterial load. Thus, there is an undoubted interest in improving PJI models to better understand the pathogenesis of this devastating clinical problem. Unfortunately, the work described in this manuscript only focuses in a single part of the animal models, namely the principles of implant design, and their data does not provide much novel insight.

Criteria for implant selection in order to reproduce the periprosthetic environment in animal models has being already described in the literature, where the most relevant existing models were presented and discussed in similar terms (implant stability, load-bearing, clinically relevant materials, separation of intra-articular and intramedullary spaces, etc.) as the authors did in the present manuscript (see reference: https://www.ncbi.nlm.nih.gov/pubmed/27707853).

In the present study, the authors provided examples of non-infected animal models, which might be suitable for infection models. However, only two of those models were mentioned in some more detail in the results section, with no deep discussion about their possible implementation to PJI models:

-Page 15 First paragraph: “The prosthesis showed great advantages in appearance, function, material science, stability, degree of matching, and so on. It can be used for reference in the design of the PJI animal model.”

In several occasions, the reference in the present study to bacterial biofilms is ambiguous or can lead to confusion:

-Page 8 Abstract’s conclusion and page 19 Conclusions: “…the surface of the prosthesis is smooth with the formation of biofilm…”

What do the authors mean by that sentence? Do they mean that the surface of the prosthesis should be smooth to allow (or avoid) biofilm formation?

-Page 16 Question 1: “…Other studies have shown that material can even form on an antibiotic-loaded bone cement [54, 55]…”

With the term “material” do the authors mean “biofilm”?

-Page 17 Question 4: “Karbysheva et al [54] supposed that the ability of bacterial adhesion and the formation of bacterial biofilm was different because of the diversity of biological materials.”

The sentence is not accurate. Karbysheva et al suggested that the microorganisms' ability to adhere and form a biofilm on different biomaterials of explanted joint prosthesis components might differs among biomaterials.

Incorrect reference to figures in pages 12-15.

The conclusion section would benefit of a brief reasoning for each mentioned criterion.

The manuscript is overall well written but it is suggested an English editing of the manuscript by a mother tongue speaker to correct spelling mistakes and improve the readability of numerous sentences.

Reviewer #2: This is a systematic review analyzing the prosthesis design used in animal models dealing with periprosthetic joint infection (PJI) following total knee arthroplasty (TKA). The review is well written and correctly follows the checklist PRISMA for systematic reviews. The aim of the study was to look at the different prosthesis designs in knee implants used in animal models of PJI. The authors have obviously the assumption that the design of the prosthesis used in the animal model is crucial for the imitation of human PJI. If this is indeed the case, the minimal infecting dose and the response to antimicrobial treatment should be closest to the human situation in the animal model using an implant which best imitates the human one. This clinically relevant part of the review is missing, because it can obviously not be extracted from the different studies. Thus, it may well be that other factors such as the pathogenesis of inoculation (exogenous during implantation vs hematogenous), the inoculum, the type of microorganism, the delay until antimicrobial therapy etc. are much more important for the precise simulation of the human situation than the design of the implant. The design of the prosthetic joint may be much more important in non-infectious situations, where factors such as function, aseptic loosening instead of susceptibility to infection are analyzed.

Specific comments

1.Abstract/Objectives. The aim of the study is to present criteria for the evaluation of a clinically representative model of PJI. Up to now, no PJI animal model perfectly simulates the design of a human TKA. Nevertheless, even with very crude imitations (see figure 2), all PJI animal models perfectly simulate the clinical situation in the sense that the minimal infecting dose is very low, the untreated PJI never spontaneously heals, and established biofilms cannot be eliminated by most antibiotics. Thus, the clinical relevance of the study objective can be contested.

2.Abstract/Results. Eight PJI animal models are presented in this review. Unfortunately, the manuscript is narrowly centered on a readership of orthopedic surgeons. Criteria which are important for Infectious Disease specialists are not reported. Among these are specifics regarding the minimal infecting dose, the spontaneous course of the infection (loosening? Sinus tract?), the expansion from PJI to concomitant osteomyelitis etc.

3.Introduction. The statement “…PJI is one of the catastrophic complications…” is clearly exaggerated, since in many cases, debridement, antibiotics and implant retention is a valuable strategy. Thus, PJI is a severe, but not a catastrophic complication.

4.Introduction. The authors state that “…the current design of PJI models is still dissatisfying.” In my view, it is correct, that none of the PJI animal model closely simulates the human situation. However, the authors should explain, why this is “dissatisfying”. Does this lead to wrong conclusions regarding pathogenetic, diagnostic or therapeutic concepts in human beings with PJI? The authors should cite such “dissatisfying” factors, in order to convince the reader that the PJI animal model should perfectly simulate the human situation.

5.Discussion/Question 1. The statement that cementless TKS is associated with higher loosening rates is not correct. Citation #48 is completely wrong. Drexler et al. state “…cemented fixation offered equivalent clinical outcomes and at least as good as, if not better, survival than uncemented fixation…”

6.Discussion/Question 1. The Charité group (citation #54) tested cobalt-chromium, titanium and polyethylene, but not bone cement.

7.Strenghts and limitations. The authors state that the strengths of their paper are “evident”. The individual strengths should be enumerated.

8.Strengths and limiations. The authors state “…most influencing factors are based on the prosthesis design, which determines the success or failure of the animal model…” However, they don’t show any examples of failed PJI animal models (cfr. #4 above). Thus, it remains unclear, whether the design is only important for non-infectious arthroplasty models, but also for PJI animal models.

Minor comment

1.Results. The authors cite 4 criteria for prosthetic design. However, they enumerate only the first two. Enumeration (3) and (4) should be added for the other two criteria.

Reviewer #3: The authors performed a systematic review aiming to summarize the prosthesis design of animal models of PJI following TKA. After a huge literature screening, they found a total of 12 studies (8 concerning models of infection, 4 non-infection models) and they summarized the different materials, animals, location and the presence/absence of cement, on order to find the best suitable prosthesis model for future animal model studies on PJIs.

Overall, the manuscript is well written and easy to read.

I have only minor comments:

- I would introduce the part concerning the different types of bacteria (gram postive vs gram negatives, inocolum) in the manuscript, not only in the tables.

- bacterial names should be written in Italic

- authors should check some spelling errors in english (i.e. tenses in the method section)

Figure2. Legend should be ameliorated. Does the figure refer to the used models in the literature of prosthesis for animal models?

- Tables: all the studies should be temporarly order

6. PLOS authors have the option to publish the peer review history of their article (what does this mean?). If published, this will include your full peer review and any attached files.

Reviewer #1: No

Reviewer #2: No

Reviewer #3: No

---

## [Author Response · Author response to Decision Letter 0]

16 Sep 2019

Dear Daniel Pérez-Prieto and the three reviewers:

Thank you for your letter and for the reviewer's comments concerning our manuscript entitled "Prosthesis Design of Animal Models of Periprosthetic Joint Infection Following Total Knee Arthroplasty: A Systematic Review". Those comments are all valuable and very helpful for revising and improving our paper. We have studied comments carefully and have made correction which we hope to meet with approval. Revised parts are underlined in red. Each point by the academic editor and reviewers was listed below and was answered sincerely by us:

Response to academic editor (Daniel Pérez-Prieto) 

Q:

Please confirm that you have included all items recommended in the PRISMA checklist including details of reasons for study exclusions in the PRISMA flowchart and number of studies excluded for each reason.

We suggest you thoroughly copyedit your manuscript for language usage, spelling, and grammar.

A:

After multiple checking, our manuscript completely meets PLOS ONE's style requirements and follows the checklist PRISMA. There truly existed some language problems, so we invited an English teacher correct the whole paper. The certificate of English editing was also attached as supporting information.

Q:

Please ensure that you refer to Figure 3 in your text as, if accepted, production will need this reference to link the reader to the figure.

A:

Thank you for correcting the mistake. I am very sorry for making such a stupid mistake. The right markers were updated.

In addition, 'citation #22' is an article published in Core journal of Peking University in China, and the original article has been uploaded as separate file and labeled ' citation #22' (The design of anatomic knee prostheses for rabbits with computer aided design).

Q:

We note that you have stated that you will provide repository information for your data at acceptance. Should your manuscript be accepted for publication, we will hold it until you provide the relevant accession numbers or DOIs necessary to access your data. If you wish to make changes to your Data Availability statement, please describe these changes in your cover letter and we will update your Data Availability statement to reflect the information you provide.

A:

Yes, our manuscript can be accepted for publication and all data in the study are fully available without restriction. It is a systematic review, so there are no relevant accession numbers or DOIs for data. I do not know whether I understand this right. 

Response to reviewer 1 

Q:

The manuscript is overall well written but it is suggested an English editing of the manuscript by a mother tongue speaker to correct spelling mistakes and improve the readability of numerous sentences.

A:

Thank you very much for your admiration and suggestion, your suggestion is very pertinent, our English language do need to be improved. Through re-reading of the full paper carefully, we have found some phrases and grammatical errors, all of the errors have been revised. We also have invited an English teacher proofread the whole paper, I hope it is more clear and accurate now of this revised paper on the English expression.

Q:

The work described in this manuscript only focuses in a single part of the animal models, namely the principles of implant design, and their data does not provide much novel insight.

Criteria for implant selection in order to reproduce the periprosthetic environment in animal models has being already described in the literature, where the most relevant existing models were presented and discussed in similar terms (implant stability, load-bearing, clinically relevant materials, separation of intra-articular and intramedullary spaces, etc.) as the authors did in the present manuscript (see reference: https://www.ncbi.nlm.nih.gov/pubmed/27707853).

A:

Thank you for pointing it out. We are so sorry not to illustrate the novelty and importance of this work clearly. We do focus on the principles of implant design of the PJI models for the following reasons. Firstly, according to the International Consensus on Orthopedic Infections in 2019, the ideal prosthesis design of a PJI model has yet to be established. Secondly, the implant design was the first step of performing PJI models, if it is far different from clinical implant, then the next steps will be affected and the results may be lack of persuasion. thirdly, there were still no special articles concerning to prosthesis design. Finally, other studies had briefly discussed implant stability, load-bearing, clinically relevant materials, separation of intra-articular and intramedullary spaces without further details. It was not enough at all. Additionally, we not only aimed at the above factors systematically but also the following problems:(1)Should bone cement be used for fixation?(2) Should one resect or preserve the posterior cruciate ligament?(3)Should only tibial replacement be performed or should tibial and femoral replacement be performed simultaneously?(4)Should a UHMWPE insert be used or not? Moreover, we also improved the PJI models which was based on non-infected animal models. Thus, in order to highlight our contributions to implant design of the PJI models, we have also enhanced the novelty and importance in the part of “introduction” and “strengths and limitations”.

Q:

Only two of those models were mentioned in some more detail in the results section, with no deep discussion about their possible implementation to PJI models：

-Page 15 First paragraph: “The prosthesis showed great advantages in appearance, function, material science, stability, degree of matching, and so on. It can be used for reference in the design of the PJI animal model.”

A:

We think the two prosthesis models are the most representative in all non-infected models, so only their details were described in the study and characteristics of the other two models are described in the table 3. Moreover, deep discussion about their possible implementation to PJI models has been added as your suggestion. 

Q:

In several occasions, the reference in the present study to bacterial biofilms is ambiguous or can lead to confusion:

-Page 8 Abstract’s conclusion and page 19 Conclusions: “…the surface of the prosthesis is smooth with the formation of biofilm…”

What do the authors mean by that sentence? Do they mean that the surface of the prosthesis should be smooth to allow (or avoid) biofilm formation?

A:

Thanks a lot for your kind remind. Our original writing does cause ambiguity. We mean that the surface of the prosthesis should be relatively smooth as like as the prosthesis in clinical practice, so it would not influence the range of motion. At the meanwhile, biofilm can form on the surface of the prosthesis. All above conditions should be met. The exact description has been presented.

Q:

-Page 16 Question 1: “…Other studies have shown that material can even form on an antibiotic-loaded bone cement [54, 55]…”

With the term “material” do the authors mean “biofilm”?

A: 

Yes, it was wrong-written.We mean “biofilm” here. The wrong word has been replaced.

Q:

-Page 17 Question 4: “Karbysheva et al [54] supposed that the ability of bacterial adhesion and the formation of bacterial biofilm was different because of the diversity of biological materials.”

The sentence is not accurate. Karbysheva et al suggested that the microorganisms' ability to adhere and form a biofilm on different biomaterials of explanted joint prosthesis components might differs among biomaterials.

A:

We were truly sorry for the wrong expression. The sentence has been modified as your suggestion.

Q:

Incorrect reference to figures in pages 12-15.

A:

We have checked it repeatedly and correct the incorrect reference.

Q:

The conclusion section would benefit of a brief reasoning for each mentioned criterion.

A:

The first four criterias have been ameliorated as your wish.

1. The surface of the prosthesis should be smooth so as not to limit knee joint ROM.

2. The surface of the prosthesis should be found the formation of biofilm.

3. The implants are composed of Ti-6Al-4V or Co-Cr-Mo alloy, with or without UHMWPE, which are similar to clinical materials.

4. The implants can bear weight and separate the intra-articular and medullary cavities for reproducing the periprosthetic environment.

Response to reviewer 2 

Q:

The review is well written and correctly follows the checklist PRISMA for systematic reviews. The authors have obviously the assumption that the design of the prosthesis used in the animal model is crucial for the imitation of human PJI. If this is indeed the case, the minimal infecting dose and the response to antimicrobial treatment should be closest to the human situation in the animal model using an implant which best imitates the human one. It may well be that other factors such as the pathogenesis of inoculation (exogenous during implantation vs hematogenous), the inoculum, the type of microorganism, the delay until antimicrobial therapy etc. are much more important for the precise simulation of the human situation than the design of the implant. 

The design of the prosthetic joint may be much more important in non-infectious situations, where factors such as function, aseptic loosening instead of susceptibility to infection are analyzed.

A:

We greatly appreciate both your help and that of the referees concerning improvement to this maniscript. 

We also agree that other factors are important. Nevertheless, the design of the implant is also as important as other factors, because the former is the the first step of performing PJI models. Most influencing factors were based on it. If the implant design can best imitate the human ones, other factors will be more persuasive. 

In my point of view, the factors, such as function, aseptic loosening, are not only important in non-infectious situations, but also in infectious ones. For examples, The instability of the prosthesis will not only affect the bone growth, resulting in loosening of the prosthesis, but it will also affect the adhesion of bacteria due to the unstable shear force.

The importance and necessity of the prosthesis design may be not obvious enough in the current study, so we have tried our best to highlight our topic.

Q:

Abstract/Objectives. Up to now, no PJI animal model perfectly simulates the design of a human TKA. Nevertheless, even with very crude imitations (see figure 2), all PJI animal models perfectly simulate the clinical situation in the sense that the minimal infecting dose is very low, the untreated PJI never spontaneously heals, and established biofilms cannot be eliminated by most antibiotics. Thus, the clinical relevance of the study objective can be contested.

A:

Thanks for your kind remind. We think the clinical relevance of the study objective was not contested for the following reasons. As described in the International Consensus on Orthopedic Infections in 2019, it is conceivable that a clinically representative animal model of PJI could improve our understanding of the pathogenesis of PJI and consequently lead to novel strategies for PJI, prevention and treatment. Although the minimal infecting dose was low in PJI models in figure 2, they all cannot reproduce the periprosthetic environment. Additionally, the pathogenesis of PJI is also different in different environment, such as the time and feature of biofilm formation. 

Q:

Abstract/Results. The manuscript is narrowly centered on a readership of orthopedic surgeons. Criteria which are important for Infectious Disease specialists are not reported. Among these are specifics regarding the minimal infecting dose, the spontaneous course of the infection (loosening? Sinus tract?), the expansion from PJI to concomitant osteomyelitis etc.

A:

We agree that other criteria, such as the minimal infecting dose and the spontaneous course of the infection and so on, are important for PJI. However, we aimed at evaluating the prosthesis design.The reason why we chose this topic was due to the dissatisfying prosthesis design in PJI models yet. Many studies had highlighted its importance in establishing PJI models, which had also been comed up with in the International Consensus on Orthopedic Infections in 2019. It was vital for recreating the periprosthetic environment. Compared with spontaneous course of the infection, the expansion from PJI to concomitant osteomyelitis or any other pathologic process, the formation of biofilm under electron microscope is the most intuitive and precise way to establish PJI models successfully. Therefore, We think the manuscript is not a superficial issue but a profound one.

Q:

Introduction. The statement “…PJI is one of the catastrophic complications…” is clearly exaggerated, since in many cases, debridement, antibiotics and implant retention is a valuable strategy. Thus, PJI is a severe, but not a catastrophic complication.

A:

This expression is not accurate enough and the wrong word has been replaced.

Q:

Introduction. The authors state that “…the current design of PJI models is still dissatisfying.” In my view, it is correct, that none of the PJI animal model closely simulates the human situation. However, the authors should explain, why this is “dissatisfying”. Does this lead to wrong conclusions regarding pathogenetic, diagnostic or therapeutic concepts in human beings with PJI? The authors should cite such “dissatisfying” factors, in order to convince the reader that the PJI animal model should perfectly simulate the human situation.

A:

In theory, we believed that the “dissatisfying” factors would lead to wrong conclusions as many studies described. The “dissatisfying” factors included the lack of weight bearing, poor matching, only partial replacement, weak stability, inconsistent relevantly clinical material, rough surface, non-anatomical appearance, and so on, which had been written and cited in the present study in page 3. Each factor was discussed in details according to the included studies. In addition, we have also complemented the reason as your suggestion. As far as I am concerned, only by imitating the periprosthetic environment in the human body to the greatest extent can we create the most clinically representative model of PJI.

Q:

Discussion/Question 1. The statement that cementless TKS is associated with higher loosening rates is not correct. Citation #48 is completely wrong. Drexler et al. state “…cemented fixation offered equivalent clinical outcomes and at least as good as, if not better, survival than uncemented fixation…”

A:

We completely agree with your opinion. Cementless TKA showed similar or even better mid-long-term survivorship than cement TKA, but the former also showed lower initial stability than the latter, which is important for the formation of biofilm. The reference 48 has been deleted and replaced by another reference. (Crook PD, Owen JR, Hess SR, Al-Humadi SM, Wayne JS, Jiranek WAJTJoa. Initial stability of cemented vs cementless tibial components under cyclic load. 2017;32(8):2556-62. doi: 10.1016/j.arth.2017.03.039. PMID: 28433426.)

In addition, this vague sentence has been corrected according to your suggestion.

Q:

Discussion/Question 1. The Charité group (citation #54) tested cobalt-chromium, titanium and polyethylene, but not bone cement.

A:

I have replaced it with another citation. 

Q:

Strenghts and limitations. The authors state that the strengths of their paper are “evident”. The individual strengths should be enumerated.

A:

Yes, we have added the strengths of their paper in the part of “Strenghts and limitations”. This is the first study concerning to the principles of implant design of the PJI models specially. We have not only systematically discussed implant stability, load-bearing, clinically relevant materials, separation of intra-articular and intramedullary spaces, but also the problems of fixation, posterior cruciate ligament, femoral replacement and UHMWPE insert.

Q:

Strengths and limiations. The authors state “…most influencing factors are based on the prosthesis design, which determines the success or failure of the animal model…” However, they don’t show any examples of failed PJI animal models (cfr. #4 above). Thus, it remains unclear, whether the design is only important for non-infectious arthroplasty models, but also for PJI animal models.

A:

Up to now, A number of literatures had explored the effects of different concentrations and different artificial materials on biofilms. However, there was no control test on comparing the influence of different prosthesis design on the PJI model, and few studies reported the example of failure model, which is worthy of further study. The expression may be not accurate enough. Thus, we have deleted the sentence “which determines the success or failure of the animal model” and changed the expression.

Q:

1.Results. The authors cite 4 criteria for prosthetic design. However, they enumerate only the first two. Enumeration (3) and (4) should be added for the other two criteria.

A:

The other two criteria (3) and (4) have been added in the part of “results”. (3) The animals chosen for models should have musculoskeletal and immunological system compared to human beings; (4) The bacteria, biofilm, and host immune response can be measured quantitatively.

Response to reviewer 3 

Q:

The authors performed a systematic review aiming to summarize the prosthesis design of animal models of PJI following TKA. After a huge literature screening, they found a total of 12 studies (8 concerning models of infection, 4 non-infection models) and they summarized the different materials, animals, location and the presence/absence of cement, in order to find the best suitable prosthesis model for future animal model studies on PJIs.

Overall, the manuscript is well written and easy to read.

A:

Special thanks to you for your good comments. I sincerely hope that it will become the criteria for evaluating a clinically representative model of PJI.

Q:

- I would introduce the part concerning the different types of bacteria (gram postive vs gram negatives, inocolum) in the manuscript, not only in the tables.

A:

I really agree that this is a very good and important point for establishing PJI models. However, because this study aimed at evaluating the prosthesis design of animal models of PJI and some other studies had described the different types of bacteria before, I am sorry about that this part was not written. I would be pleased to elaborate them in further research in the future. 

Q:

- bacterial names should be written in Italic

- authors should check some spelling errors in english (i.e. tenses in the method section)

A:

Font format, spelling and the tenses errors do exist. They have been all revised in the text.

Q:

Figure2. Legend should be ameliorated. 

A:

It was revised in the text.

Q:

Does the figure refer to the used models in the literature of prosthesis for animal models?

A:

Yes, the prosthesis in the figures are all used in animal models.

Q:

- Tables: all the studies should be temporarly order

A: 

The order of all the studies in table 1 and 2 has been adjusted.

We show much admiration and thank for the effort of Daniel Pérez-Prieto and three reviewers again. The comments were very useful, we hope our reply might address the confusions about the manuscript. We show best regards! Once again, thanks you very much!

Sincerely,

Yirong Zeng

---

## [Decision Letter · Decision Letter 1]

23 Sep 2019

Prosthesis Design of Animal Models of Periprosthetic Joint Infection Following Total Knee Arthroplasty: A Systematic Review

PONE-D-19-17399R1

Dear Dr. Zeng,

We are pleased to inform you that your manuscript has been judged scientifically suitable for publication and will be formally accepted for publication once it complies with all outstanding technical requirements.

With kind regards,

Daniel Pérez-Prieto, PhD

Academic Editor

PLOS ONE

Additional Editor Comments (optional):

Reviewers' comments:

Reviewer's Responses to Questions

**Comments to the Author**

1. If the authors have adequately addressed your comments raised in a previous round of review and you feel that this manuscript is now acceptable for publication, you may indicate that here to bypass the “Comments to the Author” section, enter your conflict of interest statement in the “Confidential to Editor” section, and submit your "Accept" recommendation.

Reviewer #2: All comments have been addressed

2. Is the manuscript technically sound, and do the data support the conclusions?

Reviewer #2: Yes

3. Has the statistical analysis been performed appropriately and rigorously? 

Reviewer #2: N/A

4. Have the authors made all data underlying the findings in their manuscript fully available?

Reviewer #2: Yes

5. Is the manuscript presented in an intelligible fashion and written in standard English?

Reviewer #2: Yes

6. Review Comments to the Author

Reviewer #2: In the revised version of the manuscript, the authors improved some linguistic problems and replaced references which did not support certain statements. Overall, the changes are minor but adequate. Most answers to the reviewers’ questions are defensive and do not resolve the indicated problem. Thus, the limitation remains, namely the very limited value of this manuscript for Infectious Disease clinicians. Nevertheless, this systematic review, which is methodologically correctly performed, may have a good value for orthopedic surgeons performing experimental work in the field of total knee arthroplasty.

7. PLOS authors have the option to publish the peer review history of their article (what does this mean?). If published, this will include your full peer review and any attached files.

Reviewer #2: No

---

## [Editor Report · Acceptance letter]

24 Sep 2019

PONE-D-19-17399R1 

Prosthesis Design of Animal Models of Periprosthetic Joint Infection Following Total Knee Arthroplasty: A Systematic Review 

Dear Dr. Zeng:

I am pleased to inform you that your manuscript has been deemed suitable for publication in PLOS ONE. Congratulations! Your manuscript is now with our production department. 

With kind regards,

on behalf of

Dr. Daniel Pérez-Prieto 

Academic Editor

PLOS ONE